# Theoretical and Experimental Studies of 1-Dodecyl-3-phenylquinoxalin-2(1H)-one as a Sustainable Corrosion Inhibitor for Carbon Steel in Acidic Electrolyte

Fouad Benhiba [1,2,*], Mohcine Missioui [3], Selma Lamghafri [4], Rachid Hsissou [5], Abdelkbir Bellaouchou [2], Hassan Oudda [1], Abdellatif Lamhamdi [4], Ismail Warad [6], Youssef Ramli [3] and Abdelkader Zarrouk [2]

1   Laboratory of Advanced Materials and Process Engineering, Faculty of Sciences, Ibn Tofail University, BP 242, Kenitra 14000, Morocco
2   Laboratory of Materials, Nanotechnology and Environment, Faculty of Sciences, Mohammed V University Rabat, Rabat 10090, Morocco
3   Laboratory of Medicinal Chemistry, Drug Sciences Research Center, Faculty of Medicine and Pharmacy, Mohammed V University Rabat, Rabat 10090, Morocco
4   Laboratory of Applied Sciences, National School of Applied Sciences (ENSAH), Abdelmalek Essaadi University, Tetouan 93000, Morocco
5   Laboratory of Organic Chemistry, Bioorganic and Environment, Chemistry Department, Faculty of Sciences, Chouaib Doukkali University, El Jadida 24000, Morocco; r.hsissou@gmail.com
6   Department of Chemistry, AN-Najah National University, Nablus P.O. Box 7, Palestine; i.kh.warad@gmail.com
*   Correspondence: fouad.benhiba@uit.ac.ma

**Abstract:** The anti-corrosion features of 1-dodecyl-3-phenylquinoxalin-2(1H)-one (QO12) for carbon steel CS were evaluated in a 1 M HCl solution using potentiodynamic polarization (PDP), electrochemical impedance (EIS) and UV-visible spectroscopy, and scanning electron microscopy (SEM), as well as quantum-chemical methods. The inhibition performance achieves a maximum of 95.33% at 0.001 M. The PDP study revealed that QO12 acts with the character of a mixed-type inhibitor. The EISs mention that the process of corrosion for CS is essentially predominated by the transfer-of-charge mechanism. Moreover, quinoxalinone adsorption follows the Langmuir adsorption isotherm. SEM snapshots show no deterioration after the contribution of QO12 compared to the reference electrolyte. Theoretical calculations suggest that the envisaged inhibitor presents a perfect arrangement capacity through the structure of quinoxalinone.

**Keywords:** quinoxalinone; corrosion inhibition; CS; PDP/EIS; quantum-chemical

## 1. Introduction

Due to its high resistance, high durability, and fairly low cost, CS remains one of the most frequently used materials in the manufacturing industries [1–4]. It is well known that the pickling process is used in the field of metal surface treatment, which involves the need to use an acid solution to reduce scale and impurities on the steel surface. Indeed, during the dissolution of the surface layer of the steel, the acidic solution can lead to the production of iron salts which can be easily oxidized, leading to corrosion [5–7]. Hydrochloric acid (HCl) belongs to the class of highly aggressive acids and can cause extensive damage to metallic equipment. This acid causes a corrosion reaction when in contact with metals such as copper, zinc, aluminum, and CS [8]. The employment of corrosion inhibitors is a widespread and effective procedure for limiting acid penetration and preventing the corrosion of metals, especially CS, during pickling operations. They form a protective layer on the surface of the metal to inhibit the corrosive reaction between the metal and the acid, resulting in reduced corrosion [9–12]. They are extensively used throughout the processing industry thanks to their good solubility, low cost and low toxicity. Several types

of inhibitors can be envisaged, in particular organic compounds. In general, the inhibitor chosen should be relative to the acid solution of study and the type of metal tested [13]. The degree of anticorrosive efficiency of organic molecules containing heteroatoms (O, N, P, and S), π-electrons, multiple bonds, and aromatic rings is high, and they are very likely to adsorb effectively on the substrate surface [14–18].

Quinoxaline derivatives have many applications in the biological, pharmaceutical, industrial, and other fields [19,20]. In addition to these activities, these compounds also possess anticorrosion characteristics; recently, our team has published several research studies to find new inhibitors, derivative of quinoxaline, to improve the inhibitory efficiency of mild steel in acidic media [19–21].

The inhibitor used in this work, 1-dodecyl-3-phenylquinoxalin-2(1H)-one (QO12), was synthesized to investigate the relevance of the C12 carbon chain on inhibitory potency using experimental and theoretical approaches.

This investigation is a study of QO12's behavior as well as an evaluation of its corrosion-inhibition efficiency for CS in 1 M HCl solution. The corrosion mitigation of QO12 was evaluated via electrochemical techniques and surface characterization using scanning electron microscopy with energy-dispersive as well as UV-visible X-ray spectroscopy (SEM/EDX). The new inhibitor is used to comprehend the inhibition mechanism and to provide clarity for its successful application as a chemical corrosion inhibitor for carbon steel in HCl.

Further, to further the aim of better knowing the mode of its inhibitory action on metallic surfaces, global quantum descriptor calculations using DFT and molecular dynamics (MD) simulation were employed to provide more comprehensive understanding of the experimental results [10,19–21].

## 2. Experimental

### 2.1. Synthesized Inhibitor

Quantities of 3-Phenylquinoxalin-2-one (4.5 mmol), potassium carbonate (5.85 mmol) and tetrakis (n-butyl) ammonium bromide (0.5 mmol) in DMF (20 mL) were added to 1-bromododecane (9 mmol). Stirring was maintained at room temperature for 24 h. The crude residue was filtered and the solvent removed. The residue was extracted with water. The organic compounds were purified by column chromatography using hexane-ethyl acetate (v/v, 95/5) (Figure 1).

**Figure 1.** Synthesis procedure of the quinoxalinone derivative QO12.

Yield: 72%, mp (°C) = 120–122,

$^1$H RMN (300 MHz, CDCl$_3$) δppm: 0.87 (t, 3H, CH$_3$, J = 6 Hz); 1.27–1.45 (m, 18H, CH$_2$); 1.69–1.78 (quin, 2H, N-CH$_2$-CH$_2$); 4.20 (t, 2H, NCH$_2$, J = 6 Hz); 7.32–8.30 (m, 9H, CH arom).

$^{13}$C RMN (75 MHz, CDCl$_3$) δ ppm: 13.10 (CH$_3$); 21.26, 21.70, 22.36, 22.40, 27.15, 27.18, 28.12, 28.19, 29.22, 29.70 (CH$_2$); 43.61 (N-CH$_2$); 113.51, 122.10, 128.06, 129.66, 130.24, 130.30, 131.40, (CH arom); 132.58, 132.18, 136.18, 152.51 (Cq); 153.01 (C = O).

Table 1 contains information on the anticorrosive properties of some important quinoxaline derivatives in 1 M hydrochloric acid for carbon steel.

**Table 1.** Anticorrosive properties of some important quinoxaline derivatives in 1 M hydrochloric acid for CS.

| Quinoxaline Derivatives | Anticorrosive Properties (%) | References |
|---|---|---|
|  1-[4-acetyl-2-(4 chlorophenyl)quinoxalin-1(4H)-yl]acetone (Q1) | 95.80 | [22] |
|  3,7-dimethyl-1-(prop-2-yn-1-yl)quinoxalin-2(1H)–one (Q2) | 94.90 | [23] |
|  2-(2,4-dichlorophenyl) -6-Nitro-1,4-dihydroquinoxaline(Q3) | 88.00 | [24] |
|  Carbon chain:C12   Quinoxalinone | 95.33 | Present work |

### 2.2. CS and HCl Prepared

The aggressive 1 M HCl electrolytes were synthesized by diluting analytical grade 37 percent HCl with distillate water. The corrosion assays were carried out in 1 mol/L HCl electrolyte in the absence and presence of quinoxalinone at concentrations ranging from $10^{-3}$ to $10^{-6}$ M.

The CS compositions with Mass % are collected in Table 2.

**Table 2.** Percentage in mass of the CS compositions.

| Alloys | C | Si | S | Cu | Mn | Cr | Co | Ti | Ni | Fe |
|---|---|---|---|---|---|---|---|---|---|---|
| Percentage in mass | 0.370 | 0.230 | 0.016 | 0.160 | 0.680 | 0.077 | 0.009 | 0.011 | 0.059 | Rest |

The surface of CS (1 cm$^2$) was subjected to polishing with SiC abrasive papers of various sizes, between 100 and 2000. This operation helped to remove irregularities and roughness from the surface, which might affect the accuracy of electrochemical measurements. Once polished, the surface is then degreased to remove any oils and fats that may be present. This usually requires the use of a solvent such as acetone, which can dissolve and remove organic contaminants. Lastly, the surface should be rinsed with distilled water in order to eliminate all traces of contaminants.

### 2.3. Experimental

All electrochemical experiments were performed operating a PGZ 301 potentiostat (Radiometer Analytical—Hach Company, Loveland, CO, USA) piloted by a computer using the Voltamaster 4 program and a thermostatically regulated double-walled cylindrical glass cell with three electrodes. The latter contains CS as the working electrode, a platinum record as the counter-electrode, and saturated calomel as the reference electrode (SCE). All experiments were carried out at 303 K. Prior to starting electrochemical tests, the CS was dipped in the electrolytes for 30 min to achieve the permanent open circuit voltage ($E_{OCP}$). At the $E_{OCP}$, EIS measurements were measured at the $E_{OCP}$ by applying a frequency from $10^5$ to $10^{-2}$ Hz with amplitude of 10 mV. Moreover, PDPs were operated from $-0.8$ to $-0.1$ V with $5 \times 10^{-1}$ mV/s as the scanning rate.

The $\eta_{PDP}$ (%) and $\eta_{EIS}$ (%) were determined according to Equations (1) and (2) [25,26].

$$\eta_{EIS}(\%) = \left( \frac{R_P - R_P^0}{R_P} \right) \times 100 \tag{1}$$

where $R_P^0$ and $R_P$ stand for the polarization resistances of an unfettered electrolyte with and without QO12, respectively.

$$\eta_{PDP}(\%) = \left( \frac{i_{corr}^\circ - i_{corr}}{i_{corr}^\circ} \right) \times 100 \tag{2}$$

where $i_{corr}^0$ and $i_{corr}$ stand for the corrosion current densities of an unfettered electrolyte with and without QO12, respectively.

Due to the unavailability of an inhibitor, the findings previously documented by our group for the PDP and EIS techniques were exploited with respect to the effect of temperature and concentration, since we worked under the same conditions [27,28].

All measurements were repeated three times for each experimental condition to ensure the reliability and reproducibility of the results, and the average values were noted.

### 2.4. SEM/EDX and UV-Visible Analysis

The CS metal supports were stored in the studied electrolyte without and with $10^{-3}$ M of the QO12 for 24 h at 303 K. After cleaning and drying, the CS surface was tested with a QUARTRO S-FEG (Thermo Fisher, Waltham, MA, USA) as a scientific instrument for scanning electron microscopy with energy-dispersive X-ray spectroscopy (SEM/EDX)

The UV-visible spectra of $10^{-3}$ M of the QO12 were recorded before and after 24 h immersion of the CS in the HCl solution using a Jenway UV-visible spectrophotometer (Thermo Fisher, Waltham, MA, USA).

### 2.5. DFT and MD Simulation Procedure

The optimization of the QO12 occurs using a DFT-B3LYP [29] with the 6-311++G(d,p) basis set in aqueous solution using the PCM solvation model. The calculation of frequencies

was made to ensure that the structure has a minimum of potential energy surfaces without imaginary (negative) frequencies. All calculations were performed via the Gauusian-09 program package [30,31]. The energies $E_{HO}$ and $E_{LU}$ of the QO12 optimized structure were used to calculate the energy gap $\Delta E gap = E_{LUMO} - E_{HOMO}$. The most important global quantum descriptors (GQD) were estimated using the vertical ionization potential ($I_v$) and vertical electron affinity ($A_v$) [29].

Molecular dynamics "MD" simulations were used to investigate the interaction of QO12 with Fe (110) systems. Similarly to our previous work [32,33], the Forcite module in Materials Studio/8 was used to model the interactions of the QO12/Fe(110)system using a simulation box ($27.30 * 27.30 * 37.13 \ \text{Å}^3$). The constructed simulation box was emptied by $33 \ \text{Å}^3$. This vacuum was filled by 500 $H_2O$, 5 $H_3O^+$, 5 $Cl^-$, and QO12. The Andersen thermostat regulated the temperature of the modelled system to 303 K in the NVT ensemble with a simulation time of 1000 ps and 1.0 fs, all within the COMPASS force field [34].

## 3. Results and Discussion

### 3.1. Potentiodynamic Polarization

Figure 2 clearly highlights that the anodic and cathodic curves shift to lower $i_{corr}$, indicating that the addition of QO12 molecules clearly inhibits the corrosion process as a consequence of its adsorption and that this effect grows stronger with increasing inhibitor concentration. Furthermore, we can see that the decrease in the anodic arms is much slower than the downtrend of the cathodic arms. Therefore, it can be shown that the protective effect of QO12 for the cathodic reaction is significantly greater than that for the anodic reaction. The parallel cathodic and anodic arms suggest that the quinolin-8-ols did not alter the corrosion mechanism of the medium under study and that the process fundamentally proceeded through a charge-transfer mechanism.

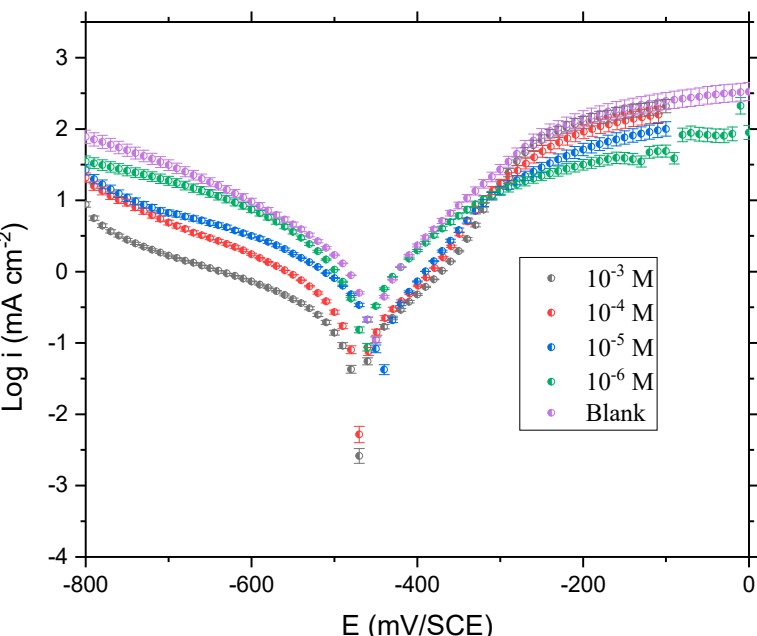

**Figure 2.** Curves of the PDP of CS surface in 1 M HCl with and without QO12 at various concentrations at 303 K.

Table 3 recapitulates the extrapolated values of PDP parameters, such as $E_{corr}$, $i_{corr}$, and anodic ($\beta a$) and cathodic ($\beta c$) Tafel slopes.

The findings in Table 3 show that the $i_{corr}$ value decreases with increasing QO12 concentrations; the $i_{corr}$ value is sufficiently high for the blank solution (1104.3 $\mu A/cm^2$) and is lowered (51.55 $\mu A/cm^2$) at $10^{-3}$ M QO12 as given in Table 3. In addition, the $E_{corr}$ of the uninhibited electrolyte was $-456.3$ (mV/SCE), and the variation in corrosion potential

is between 20 and 85.00 mV/ECS. This confirms the dual as well as mixed-action character of the inhibitor [35].

**Table 3.** Polarization parameters for CS in 1 M HCl only, containing different concentrations of QO12 at 303 K.

| Inhibitor | C. (M) | $-E_{corr}$ (mV/SCE) | $i_{corr}$ (µA/cm²) | $-\beta c$ (mV/dec) | $\beta a$ (mV/dec) | IE (%) |
|---|---|---|---|---|---|---|
| HCl | 1 | 456.3 ± 6 | 1104.00 ± 4.9 | 112.8 ± 6 | 155.4 ± 5 | - |
| | $10^{-3}$ | 435.7 ± 5 | 51.55 ± 4.4 | 153.8 ± 7 | 51.8 ± 5 | 95.33 |
| QO12 | $10^{-4}$ | 452.3 ± 5 | 72.89 ± 5.1 | 145.6 ± 6 | 54.8 ± 6 | 93.39 |
| | $10^5$ | 465.1 ± 4 | 188.51 ± 4.5 | 73.4 ± 4 | 59.6 ± 5 | 82.92 |
| | $10^{-6}$ | 463.9 ± 7 | 483.92 ± 6.2 | 92.6 ± 4 | 107.2 ± 7 | 56.17 |

The $\beta a$ values suggest that QO12 adsorbs onto the metal support and blocks the anodic sites on the CS surface without changing the mechanism of the anodic oxidation [36]. In addition, the $\beta c$ values change with QO12, suggesting that the presence of the examined molecule provokes a modification of the hydrogen reduction. These results are attributable to the barrier effect on the CS surface [37]. Bockris and Srinivasan report that this process may be due to the decrease in the cathodic transfer factor and may also be ascribed to the thickening of the electrical double layer due to the adsorption of inhibitor molecules [38].

### 3.2. EIS Investigation

EIS measurements were investigated understand the mechanism involved in corrosion processes, and to characterize films formed on CS substrates. The EIS technique allowed us to assess the corrosion inhibition by the investigated inhibitor under various concentrations. Figure 3 shows the Nyquist diagrams of CS in 1 M HCl solution with and without the QO12. The analysis of curves indicates one capacitive semi-loop in the zone of high frequencies (HF). This indicates that the process of corrosion for CS is predominated by the charge-transfer process [39,40]. After addition the studied inhibitor to the corrosive medium, the dissolution of the metal does not allow for modification of this mechanism since the all forms of loops are the same [41,42]. Additionally, it can be observed that the size of the capacitive semicircle increased with the addition of QO12 and they are bigger than that in the blank solution (1 M HCl only), which confirms the anticorrosive power of QO12 on CS in 1 M HCl. In addition, the heterogeneity of the CS area leads to semi-circles that are not perfect [43].

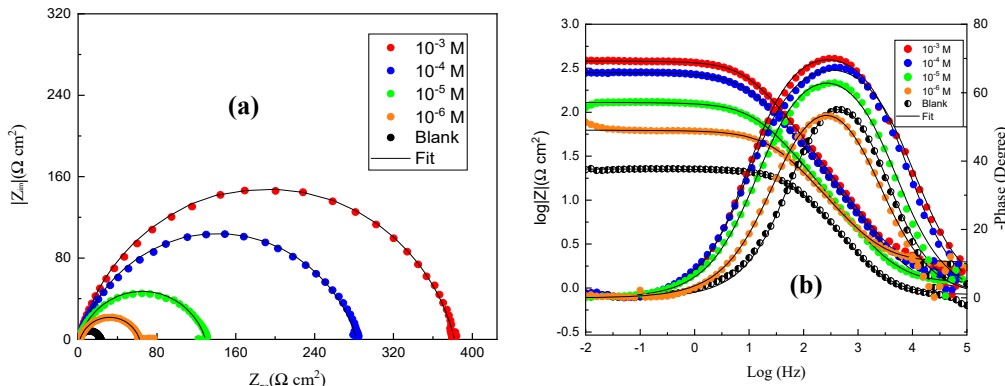

**Figure 3.** EIS (**a**), Phase, and Bode (**b**) diagrams of CS in the presence of QO12 and without at 303 K.

From the Bode curves, it can be noticed that the magnitude values of impedance increase progressively as the inhibitor concentration increases in the low-frequency zone. The impedance result is higher than that obtained with 1 M HCl, indicating the development of a protective layer on the surface of the CS. At high frequency, the values of log|Z| tend to 0; this is attributed to the electrolyte resistance [12,39]. However, a single phase

angle is observed in the plot, meaning one time constant; this indicates that the corrosion process is mainly controlled by the charge-transfer resistance, which is less than 90° for all frequencies, confirming the non-ideal comportment of the system [41].

The different processes of the CS/electrolyte is illustrated by the equivalent electrical circuit (EEC) (Figure 4); using this model, we were able to fit the results to derive the EIS parameters required to comprehend the QO12 under study [44].

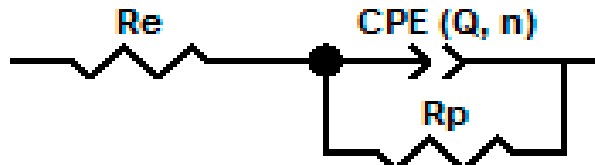

**Figure 4.** EEC for the CS/electrolyte interface during QO12 adsorption.

The equivalent circuit depicted in Figure 4 was chosen because it gave the chi squared values $\chi^2 \approx 0.001$ (Table 4).

**Table 4.** Impedance parameters of CS in 1 M HCl in the presence and in absence of QO12 at 303 K.

| Inh. | $C$ (M) | $R_s$ ($\Omega$ cm$^2$) | $R_p$ ($\Omega$ cm$^2$) | $C_{dl}$ ($\mu$F cm$^{-2}$) | $Q$ ($\mu$F s$^{n-1}$ cm$^{-2}$) | $n_{dl}$ | $\eta_{EIS}$ (%) | $\chi^2 \times 10^{-3}$ |
|---|---|---|---|---|---|---|---|---|
| HCl | 1 | 0.83 $\pm$ 1.01 | 21.57 $\pm$ 0.56 | 120.4 | 293.90 $\pm$ 2.35 | 0.845 $\pm$ 0.003 | - | 2.0 |
| QO12 | $10^{-3}$ | 1.26 $\pm$ 1.00 | 380.60 $\pm$ 0.52 | 43.9 | 83.31 $\pm$ 2.12 | 0.836 $\pm$ 0.002 | 94.3 | 1.1 |
| | $10^{-4}$ | 0.95 $\pm$ 1.19 | 281.90 $\pm$ 0.50 | 57.9 | 127.06 $\pm$ 2.32 | 0.809 $\pm$ 0.001 | 92.3 | 1.3 |
| | $10^{-5}$ | 1.14 $\pm$ 1.09 | 128.20 $\pm$ 0.53 | 87.1 | 206.06 $\pm$ 2.28 | 0.808 $\pm$ 0.003 | 83.1 | 0.3 |
| | $10^{-6}$ | 1.94 $\pm$ 1.29 | 60.42 $\pm$ 0.58 | 89.3 | 263.34 $\pm$ 2.37 | 0.793 $\pm$ 0.004 | 64.2 | 0.3 |

The $C_{dl}$ value was calculated using Equation (3) below [5]:

$$C_{dl} = \sqrt[n]{Q \cdot R_p^{1-n}} \qquad (3)$$

where $Q$ stands for the CPE modulus; $n$ ($n_{dl}$) is the deviation index ($-1 \leq n \leq 1$), which is related to the heterogeneity of the electrode surface.

The electrochemical parameters ($R_s$, $R_p$, $C_{dl}$, $Q$, and $n_{dl}$) are listed in Table 4.

Based on Table 4, it can be seen that the $R_p$ values rise with rising QO12 concentration. The highest $R_p$ value (380.6 $\Omega$ cm$^2$) has been obtained at $10^{-3}$ M, indicating the adsorption of QO12 and the development of a protective layer at the CS/solution interface [45].

Whilst the $C_{dl}$ value decreases with the addition of QO12, it changes from 120.4 $\mu$F cm$^{-2}$ for 1 M HCl medium only to 43.91 $\mu$F cm$^{-2}$ for $10^{-3}$ M of QO12. This diminution of $C_{dl}$ values with the addition of the molecule can be explained by the rises of the dielectric constant and/or an increasing of the thickness of the electrical double layer. This suggests that the inhibitor examined is capable of adsorbing onto the surface of the CS [46]. Further, the CPE constant values decrease, indicating that the organic compound interacted with the CS surface by occupying the exposed active centers of the metallic substrates [47]. Additionally, it can be noticed from the obtained results that the value of $n_{dl}$ with QO12 in the solution was higher than that of the blank; this is interpreted as a reduction in the roughness of the surface [39]. The values of inhibition power show that the highest inhibitory performance is observed at the optimum concentration ($10^{-3}$ M) and reach a maximum 94.30%, indicating that the investigated inhibitor is very powerful in corrosion inhibition for metallic substrates in the medium studied.

### 3.3. Temperature Impact and Thermodynamic Indices

According to the literature, corrosion rates generally decrease as temperature increases. With the aim of better apprehending the functioning of the studied inhibitor and its effectiveness at high temperature, the polarization curves were investigated at diverse temperatures (303 to 333 K) in a 1 M HCl solution, uninhibited and inhibited with $10^{-3}$ M of QO12 inhibitor. Figure 5 displays the polarization curves uninhibited and inhibited with $10^{-3}$ M of QO12 at temperatures studied in the 1 M HCl medium. The PDP analysis indicates the increases in $i_{corr}$ with increasing temperature (Table 5). These results are due to the desorption of the QO12 from the surface of the CS [20]. The PDP parameters as well as the values of the inhibitory efficiency of temperatures studied in 1 M HCl medium, uninhibited and inhibited with at $10^{-3}$ M of QO12, are listed in Table 5.

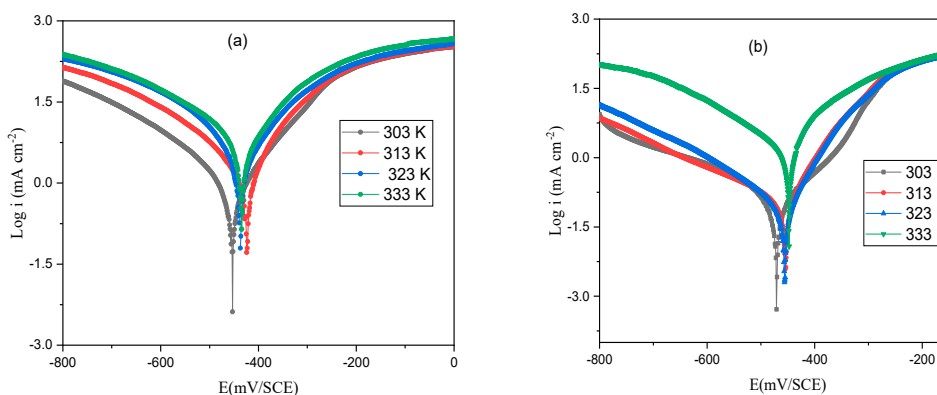

**Figure 5.** PDP curves of CS in the temperature range in 1 M HCl without (**a**) and with (**b**) the $10^{-3}$ M of QO12.

**Table 5.** PDP parameters for metallic substrate in the medium, uninhibited and inhibited with $10^{-3}$ M of QO12 at 303, 313, 323, and 333 K.

| Inhibitor | Temp. (K) | $-E_{corr}$ (mV vs. SCE) | $i_{corr}$ ($\mu$A/cm$^2$) | $E_{PDP}$ (%) |
|---|---|---|---|---|
| Blank | 303 | $456.3 \pm 6$ | $1104.1 \pm 4.9$ | - |
| | 313 | $423.5 \pm 9$ | $1477.4 \pm 7.8$ | - |
| | 323 | $436.3 \pm 7$ | $2254.0 \pm 10.2$ | - |
| | 333 | $433.3 \pm 5$ | $3944.9 \pm 12.2$ | - |
| QO12 | 303 | $435.7 \pm 5.0$ | $51.5 \pm 4.4$ | 95.3 |
| | 313 | $455.7 \pm 5.4$ | $107.4 \pm 5.6$ | 92.7 |
| | 323 | $454.0 \pm 5.3$ | $198.8 \pm 5.9$ | 91.1 |
| | 333 | $447.7 \pm 5.6$ | $382.8 \pm 5.1$ | 90.2 |

As evidenced by the results in Table 5, the increases in temperature induce increases in $i_{corr}$ values. Further, the inhibitory efficiencies values for $10^{-3}$ M of QO12 are highly decreased with increasing temperature from 95.3% (303 K) to 90.2% (333 K). This results can be explained by desorption of the QO12 inhibitor at the CS surface. In addition, for all temperature range (from 303 to 333 K) the inhibitory efficiency is higher than 90.2% indicating the effectiveness of QO12 in high temperature [46,48].

The temperature effect provides the ability to calculate the thermodynamic parameters, such as activation energy ($E_a^*$), activation enthalpy variation ($\Delta H_a^*$), and activation entropy variation ($\Delta S_a^*$), according to the following [49]:

$$i_{corr} A exp\left(\frac{-E_a^*}{RT}\right) \tag{4}$$

$$i_{corr} = \frac{RT}{Nh} exp\left(\frac{\Delta S_a^*}{R}\right) exp\left(-\frac{\Delta H_a^*}{RT}\right) \tag{5}$$

Apprehending the processes that drive the corrosion action, the $\Delta H_a^*$ and $\Delta S_a^*$ control the temperature and the disorder difference of this effect, respectively. Figures 6 and 7 illustrate the use of these equations and Table 6 groups all the results.

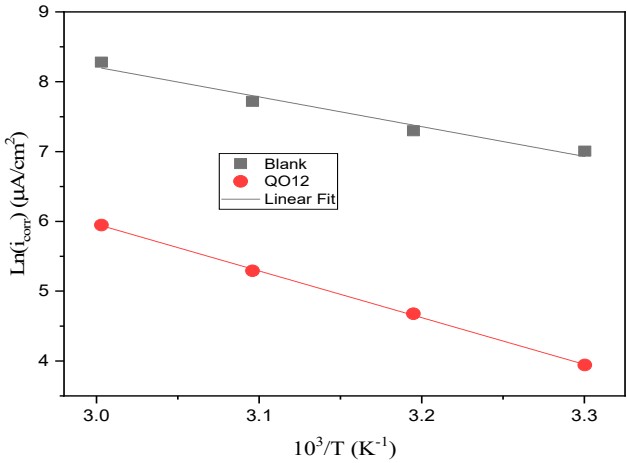

**Figure 6.** Ln ($i_{corr}$) versus 1000/T of S.C at acid medium in the absence and with the addition of the QO12.

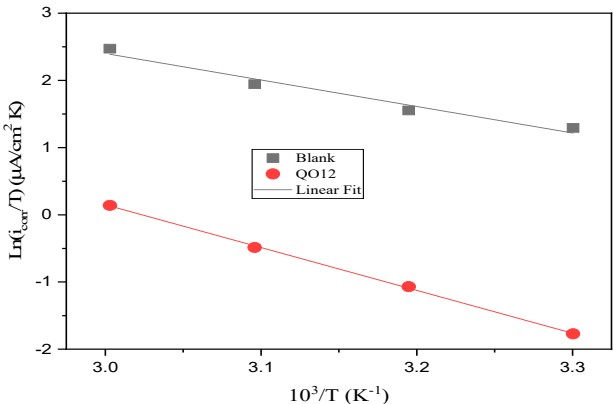

**Figure 7.** Ln ($i_{corr}$/T) versus 1000/T in blank, and addition the QO12.

**Table 6.** Activation parameter of the QO12.

| Elements | $E^*_a$ (kJ/mol) | $\Delta H^*_a$ (kJ/mol) | $\Delta S^*_a$ (J/mol K) |
|---|---|---|---|
| Blank | 35.41 | 32.77 | −81.11 |
| QO12 | 55.63 | 52.99 | −37.30 |

As data displayed in Table 6, the $E^*_a$ value of the inhibited solution (55.63 kJ/mol) is superior than that of the blank (35.42 kJ/mol); this can be accounted for by the increase in the energy barrier with the presence of the organic compound and the electrostatic adsorption action of the inhibitor on the metallic substrate [50]. Therefore, the corrosion rate is reduced. $\Delta H^*_a$ positive value conveys the endothermic reaction type of the corrosion process, while the increasing value of $\Delta H^*_a$ upon the addition of QO12 indicates a reduction in the degradation of CS. Additionally, the $\Delta S^*_a$ value for quinoxalinone is clearly higher than those found in the reference medium, implying a decrease in the disorder during the transformation of the activated reaction into an iron-compound complex in the electrolyte [51].

### 3.4. Isotherm of Adsorption

The isotherm of adsorption can supply additional information, helping to understand the mode of adsorption of the organic compound on the CS; for this reason, many trials used common isotherm models, such as the Langmuir, Temkin, Frumkin, and Freundlich, to establish which one fit the experimental data best [21]. The recovery rate $\theta$ was investigated from the derived EIS measurements. According to the adsorption isotherms, $\theta$ was related to the inhibitor concentration by the following Equations [52,53]:

$$\text{Temkin isotherm:} \exp(f \cdot \theta) = K_{ads} \cdot C \tag{6}$$

$$\text{Langmuir isotherm:} \frac{\theta}{1 - \theta} = K_{ads} \cdot C \tag{7}$$

$$\text{Frumkin isotherm:} \frac{\theta}{1 - \theta} \exp(-2f\theta) = K_{ads} \cdot C \tag{8}$$

$$\text{Freundlich isotherm:} \theta = K_{ads} \cdot C \tag{9}$$

where $K_{ads}$, $C$, and $f$ are the adsorption-desorption constant, concentration, and index of energetic inhomogeneity, respectively.

Figure 8 shows the various graphs of the Langmuir, Frumkin, Temkin, and Freundlich isotherm data. As observed in this figure, the fit of the data points for the Temkin, Frumkin, and Freundlich isotherms do not align; this is confirmed by the $R^2$ values. On the other hand, the function is linear and the correlation coefficient ($R^2$) is equal 1 for the Langmuir isotherm (Figure 8a); this shows that the adsorption of QO12 onto CS follows this isotherm.

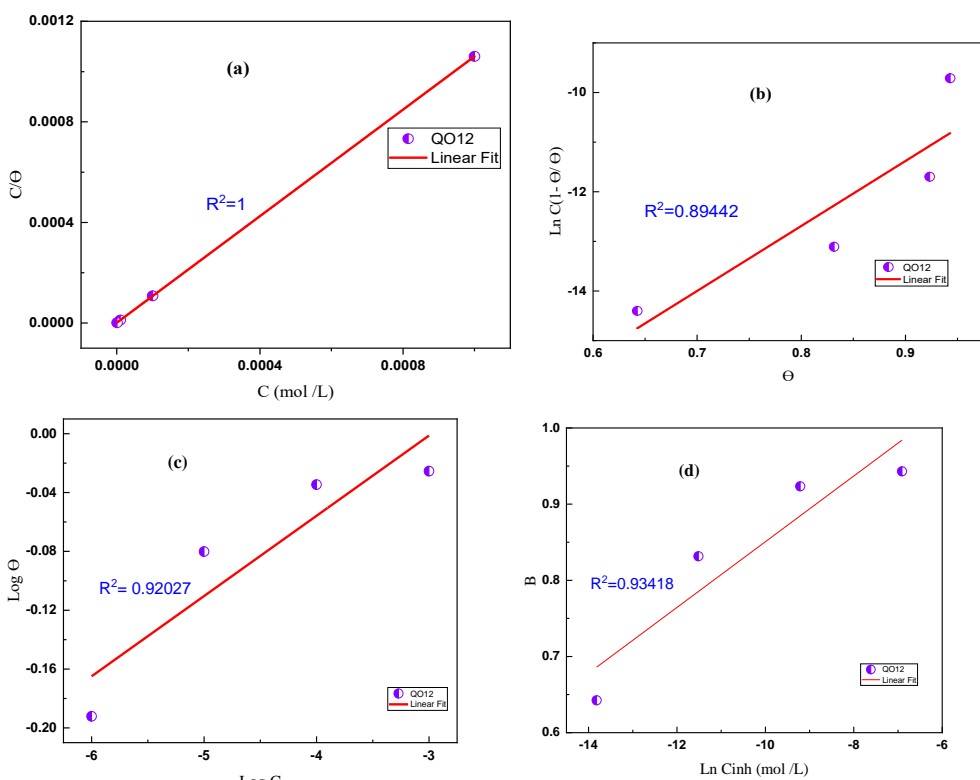

**Figure 8.** Isotherm curves for CS in 1 M HCl containing QO12 (Langmuir (**a**), Frumkin (**b**), Freundlich (**c**), and Temkin (**d**)).

The relationship between the adsorption standard free energy $\Delta G°_{ads}$ and $K_{ads}$ was calculated using the following Equation [54]:

$$\Delta G°_{ads} = -RTln(K_{ads} * C_{solvent}) \tag{10}$$

As the data show in Table 7, it can be noticed that the $\Delta G^{\circ}_{ads}$ value is equal to $-44.05$ kJ mol$^{-1}$. The fact that the value is negative implies that the adsorption phenomenon of the inhibitor in question on the metal substrate is spontaneous. According to the literature, when the $\Delta G^{\circ}_{ads}$ is about $-20$ kJ mol$^{-1}$ this indicates an electrostatic (physisorption) reaction metal/inhibitor. However, when the adsorption energy higher than $-40$ kJ mol$^{-1}$, it can be explained by charge transfer or electron sharing between the inhibitor molecule and the CS area [18]. The high value of $\Delta G^{\circ}_{ads}$ reflects the strong interaction with the CS, that is, the electron sharing taking place between the studied inhibitor and the CS by forming the coordinate bonds. Hence, it can be concluded that the investigated inhibitor is adsorbed on the CS by strong chemisorption bonds [8].

**Table 7.** Adsorption-thermodynamics magnitudes of QO12.

| Langmuir Isotherm | $R^2$ | *Slope* | $K_{ads}10^3$ (M) | $\Delta G^{\circ}_{ads}$ (kJ/mol) |
|---|---|---|---|---|
| QO12 | 1 | 1.059 | 709.486 | $-44.05$ |

*3.5. UV-Visible Spectroscopy*

In this work, an ultraviolet-visible (UV-vis) spectrum was investigated in order to study the possibility of complex formation.

Figure 9 shows the UV-vis spectra of QO12 before and after the immersion of CS at room temperature for 24 h in 1 M HCl at $10^{-3}$ M of QO12.

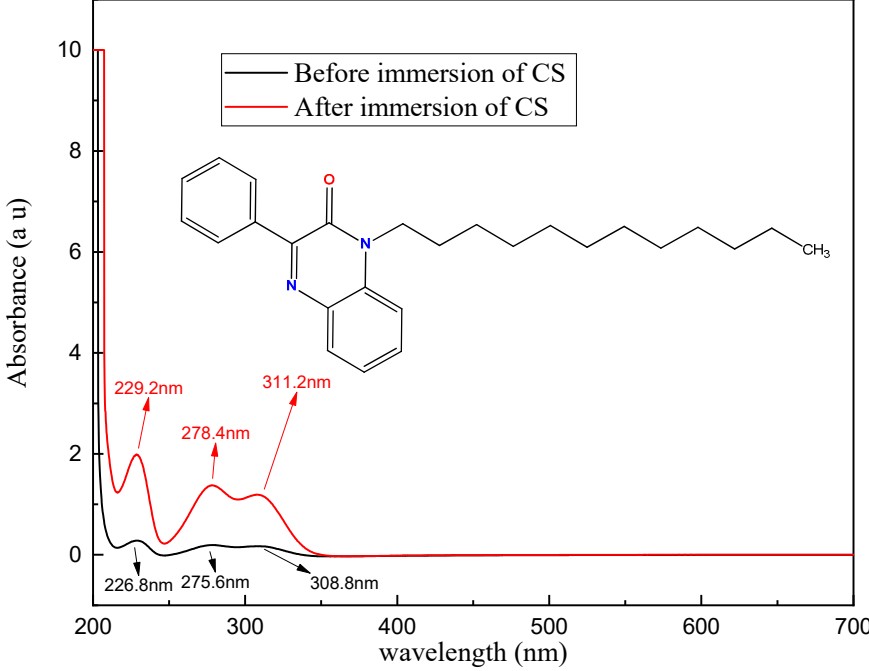

**Figure 9.** UV spectra of QO12 before and after immersion of CS at 303 K for 24 h in 1 M HCl at $10^{-3}$ M of QO12.

The absorption spectrum shows three absorption bands at wavelengths of 226.8, 275.6, and 308.8 nm before immersion, which is likely to be ascribed to n $-\pi^*$ and $\pi-\pi^*$ transitions. However, after 24 h of immersion these bands are displaced to 229.2, 278.4, and 311.2 nm, respectively.

According to the literature, if the absorbance maximum changes the position and/or values, this indicates the formation of a complex among the soluble species present in the electrolytic solution [51]; from the absorption spectra of QO12, a displacement of the wavelength has been observed, and also a diminution in the absorbance intensity values. This is usually attributed to metal–inhibitor interactions and the formation of a complex between Fe and the inhibitor.

### 3.6. SEM-EDX Investigation

Figure 10 depicts SEM observations and EDX spectra of CS only, in 1 M HCl without and with $10^{-3}$ M of the QO12. When the metal was submerged in 1 M HCl for 24 h, the surface was significantly corroded and had many quantities of corrosion products, indicating substantial corrosion (Figure 10b) [54]. However, in the presence of QO12 (Figure 10c), the surface smoothness significantly improved, showing a significant reduction in corrosion rate. This enhancement in surface morphology is attributed to the formation of a protective film on the CS surface, which is responsible for corrosion inhibition. The EDX spectrum of reference solution in Figure 10 shows Cl and O signals for the ingredients of the 1 M HCl solution, as well as iron oxide signals, indicating that corrosion products had built up on the steel surface. The EDX spectra related to the oxide of corrosion products partially disappeared after the addition of QO12, and the peak of chlorine was greatly decreased.

CS only (**a**)

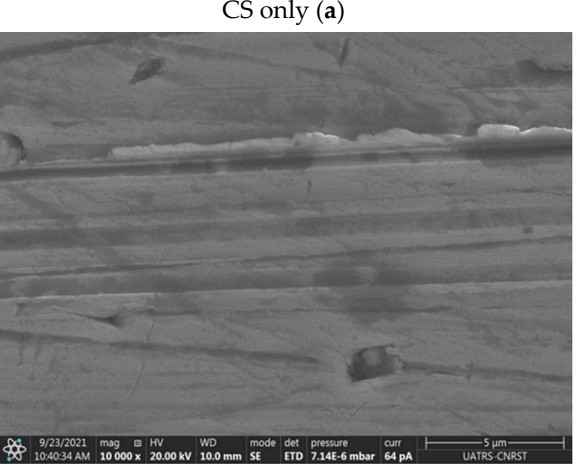

CS immersed in 1 M HCl (**b**)

CS immersed in 1 M HCl with QO12 inhibitor(**c**)

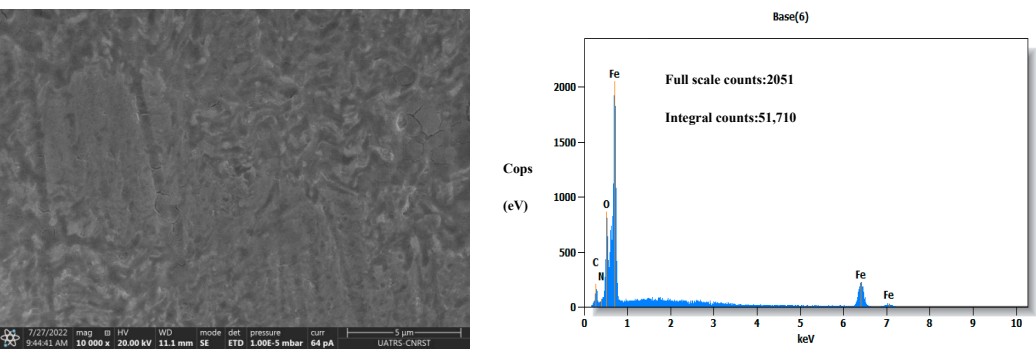

**Figure 10.** SEM-EDX for CS (**a**), in 1 M HCl only (**b**), and in 1 M HCl with 0.001 M of QO12 (**c**).

### 3.7. DFT Approaches

A set of the electronic properties, global reactivity descriptors, and local reactivity indices of the neutral form of the QO12 were calculated to examine the corrosion inhibition efficiencies and to suggest a possible corrosion-inhibition mechanism [55]. Marvin Sketch software was employed to locate the protonation site(s) [56]. Based on this, the structure of QO12 does not have a protonation site(s). To ensure the conformity of the structure, it is necessary to have the absence of imaginary frequencies using the identical level of theory. GaussView/5 software was utilized [57] to trace the structure, HOMO, and LUMO for the QO12 molecule, as illustrated in Figure 11. Close examination of the HOMO_LUMO of the QO12 molecule reveals that it is essentially concentrated over almost the entire base structure of quinoxalinone, with the exception of the joint moiety (C12). These results have been approved by the analysis of the ESP maps (Figure 12). The results obtained show the tendency of QO12 to be adsorbed on the carbon steel surface and consequently increase the corrosion inhibition efficiencies of the investigated species, which is in agreement with the provided experimental data.

**Optimized geometry**   **HOMO**   **LUMO**

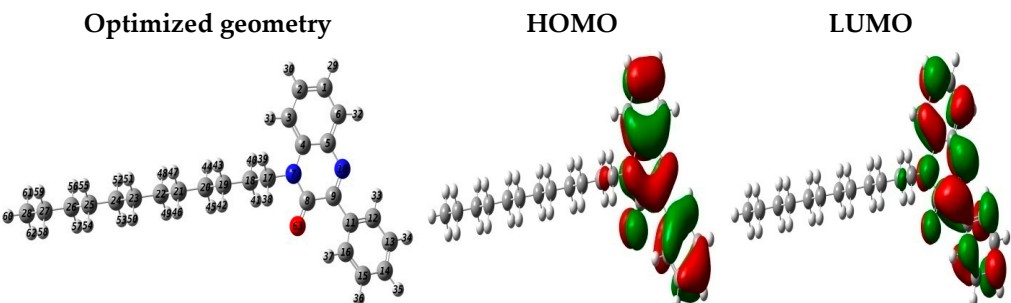

**Figure 11.** Optimized structure, HOMO, and LUMO surfaces of the neutral inhibitor QO12.

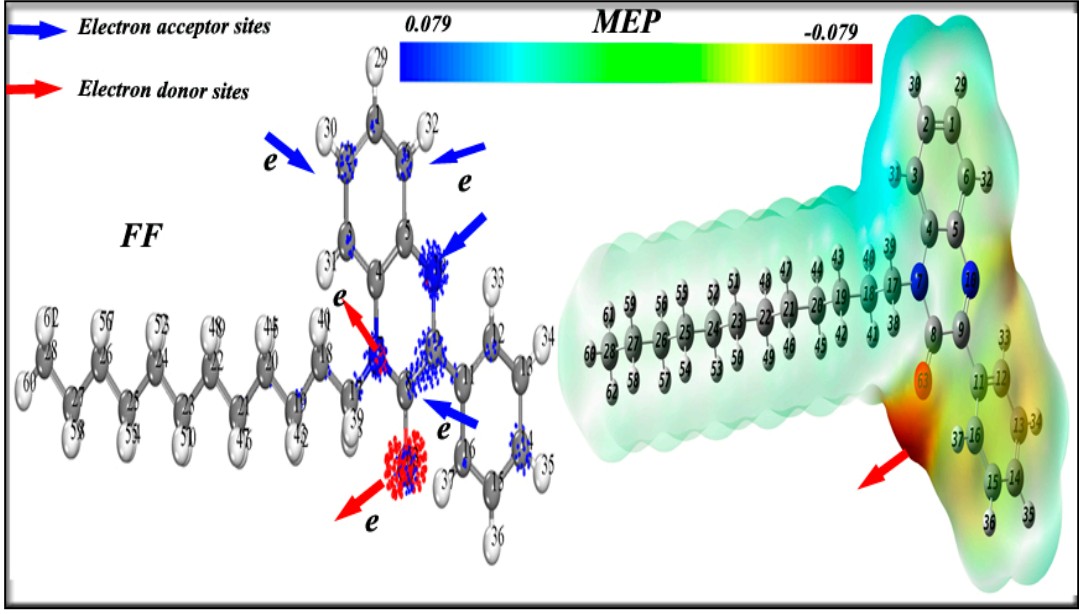

**Figure 12.** FF and MEP distributions for the QO12.

$E_{HO}$ and $E_{LU}$ values serve to analyze the donor/acceptor ratio in the molecule/metal relationship [58,59]. Indeed, an inhibitor molecule tends to be more effective against corrosion as the $E_{HO}/E_{LU}$ ratio is increased/reduced. Table 8 contains the different data from this investigation. It appears that the high value of $E_{HO}$ (−5.847 eV) and the low value of $E_{LU}$ (−1.994 eV) attest that QO12 has the ability to donate/accept electrons conveniently. $\Delta E_{gap}$ ($E_{LU} - E_{HO}$) allows us to measure the reactivity of a molecule; as

the most minimal value of this descriptor, this specific reactivity is maximal, compatible with QO12 (3.853 eV) [60,61]. When the value of hardness ($\eta$) diminishes, the reactivity of a molecule gets more prominent and interacts more successfully with the surface of a metal. Furthermore, it should be concluded from Table 8 that the values of $\eta$ and electro-negativity ($\chi$) of QO12 are acceptable, indicating the molecule may possess good anti-corrosive characteristics and can prevent the corrosion of Fe metal.

**Table 8.** Quantum chemical descriptors of the QO12 molecule (In eV).

| $E_{HO}$ | $E_{LU}$ | $\Delta Egap$ | $\chi$ | $\eta$ | $\Delta N_{110}$ | TE |
|---|---|---|---|---|---|---|
| −5.847 | −1.994 | 3.853 | 3.920 | 1.926 | 0.233 | −32,546.928 |

The fraction of electron-transferred ($\Delta N_{110}$) values provide useful information on the tendency of electrons to flow from the inhibitor molecule to the carbon steel surface ($\Delta N > 0$) or from the carbon steel to the inhibitor ($\Delta N < 0$) [62]. Inspection of the $\Delta N_{110}$ values in Table 8 indicates that the electrons can flow from the neutral form to the CS surface. Another descriptor used to reliably assess the responsiveness and stability of a molecule is the total energy (TE). From Table 8, the lowest negative value (−32,546.928 eV) suggests QO12's strong reactivity.

To study the adsorption process of the QO12 onto CS, molecular electrostatic potential (MEP) and Fukui function analysis were used to estimate the nucleophilic and electrophilic attacks caused by the most active centers of the QO12 and enhance the experimental results [63]. Figure 12 shows the 3D iso-surface of the Fukui functions and the MEP for nucleophilic ($f_k^+$) and electrophilic ($f_k^-$) attack centers. In alignment with the results of the HOMO and LUMO surface, these isosurfaces clearly indicate the active region for nucleophilic and electrophilic attacks in the investigated species.

In addition, the MEP can recognize both acceptor and donor electron sites, and this feature is reflected in several colors: red for electron donor, blue for electron acceptor, and other colors (not significant) [64].

Figure 12 makes it clear that the atoms N7, N16, and O63 are the electrophilic attack centers.

These contribute to reinforcing the importance of electron donation to acceptor locations on the Fe surface. However, the major electron acceptor sites (atoms) are located in the quinoxalinone part (blue color) in order to raise the arrangement degree of the QO12 on the CS.

### 3.8. MD Simulation Investigations

3.8.1. QO12/Fe (110) System

Computational methods, such as MD simulations, can be valuable in helping to predict and understand the adsorption pattern of a molecule onto a metal. In addition, the MD is a potent tool for exploring the behavior of molecular systems [65]. The aim of the present MD is to evaluate and understand the way in which the neutral form for the QO12 molecule operates on the iron atomic support (Fe(110)) [66,67]. Figure 13 shows the side and superior perspectives of the QO12 deposited on the Fe atomic layer together. The QO12 adsorbs using the structure of the quinoxalinone base without the carbon chain (C12) on the examined surface, which proves that the simulated molecule carries more reactive sites localized in the part filled by the FMO (HOMO_LUMO) with the iron atoms, and has great potential to adsorb onto the CS through the presence of the coordination bonds inhibitor -Fe(110).

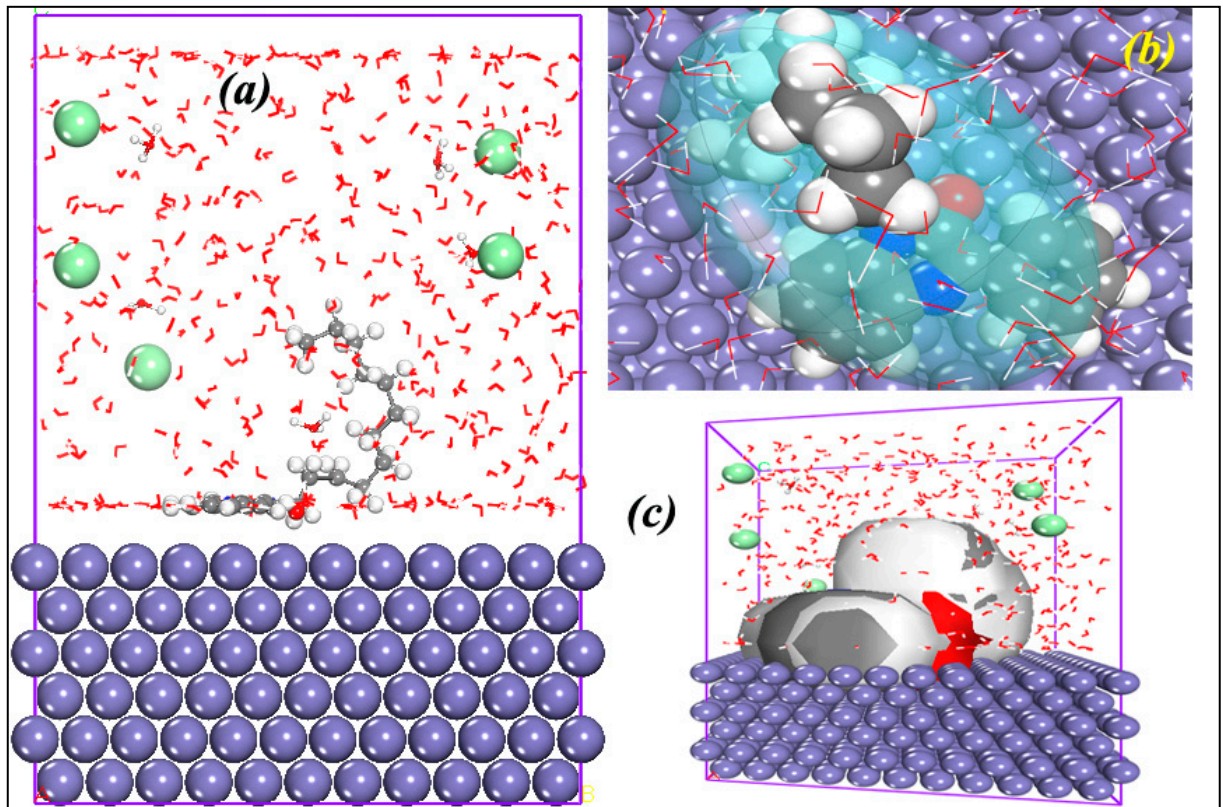

**Figure 13.** (**a**) Side, (**b**) top, (**c**) density field arrangements for the QO12/Fe (110).

The $E_{interaction}$ ($E_{inter}$) value is defined by the following equation [68]:

$$E_{inter} = -E_{(surface+solution)} - E_{QO12} + E_{total} \qquad (11)$$

The low value of $E_{inter}$ attests to the interaction of QO12 with the interacting iron atoms [69,70]. The value of this descriptor for the system is calculated; the value of the QO12/Fe(110) system is $-871.306$ kJ mol$^{-1}$, indicating a better reactivity

### 3.8.2. RDF

The radial distribution function "RDF" was entered as the main purpose of this method for assessing the adsorption suitability of the bonds between the QO12 (N7, N16, and O63) and the Fe atoms [71]. The research literature has shown that if the bond length values are shorter than 3.5 Å, adsorption of a chemical nature is most probable. Alternatively, adsorption of an electrostatic nature becomes more likely [72]. Figure 14 gives the peak data of this investigation. The numbers recorded in the first peak prove that the bond lengths of QO12 towards the Fe atoms of the first layer are shorter than 3.5 Å, informing us that QO12 is highly bonded to the metal substrate, implying improved inhibitory protection.

### 3.8.3. MSD Tool

The film inhibition potential shown by QO12 against Cl$^-$ and H$_3$O$^+$ ions was assessed with the fractional free volume (FFV).

If the inhibitor film contains considerable cavities, the dispersion of the ions is raised automatically, so the inhibition performance is automatically lowered [73].

Figure 15d shows the molecular dynamics simulation reaches the fundamental state with the curve stabilizing around the average temperature of 303 K.

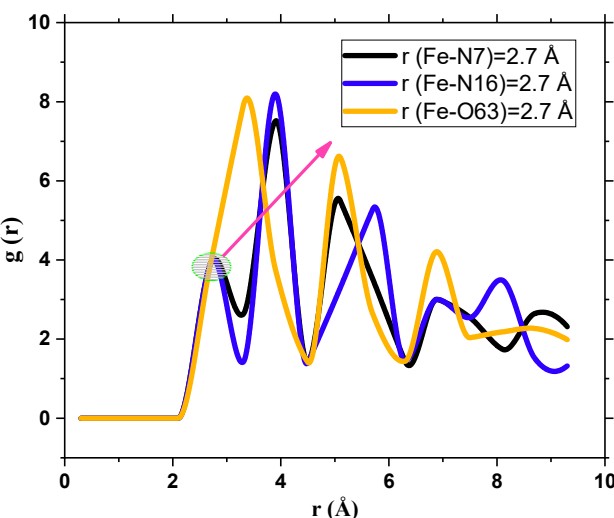

**Figure 14.** Analysis of RDF for QO12/Fe (110).

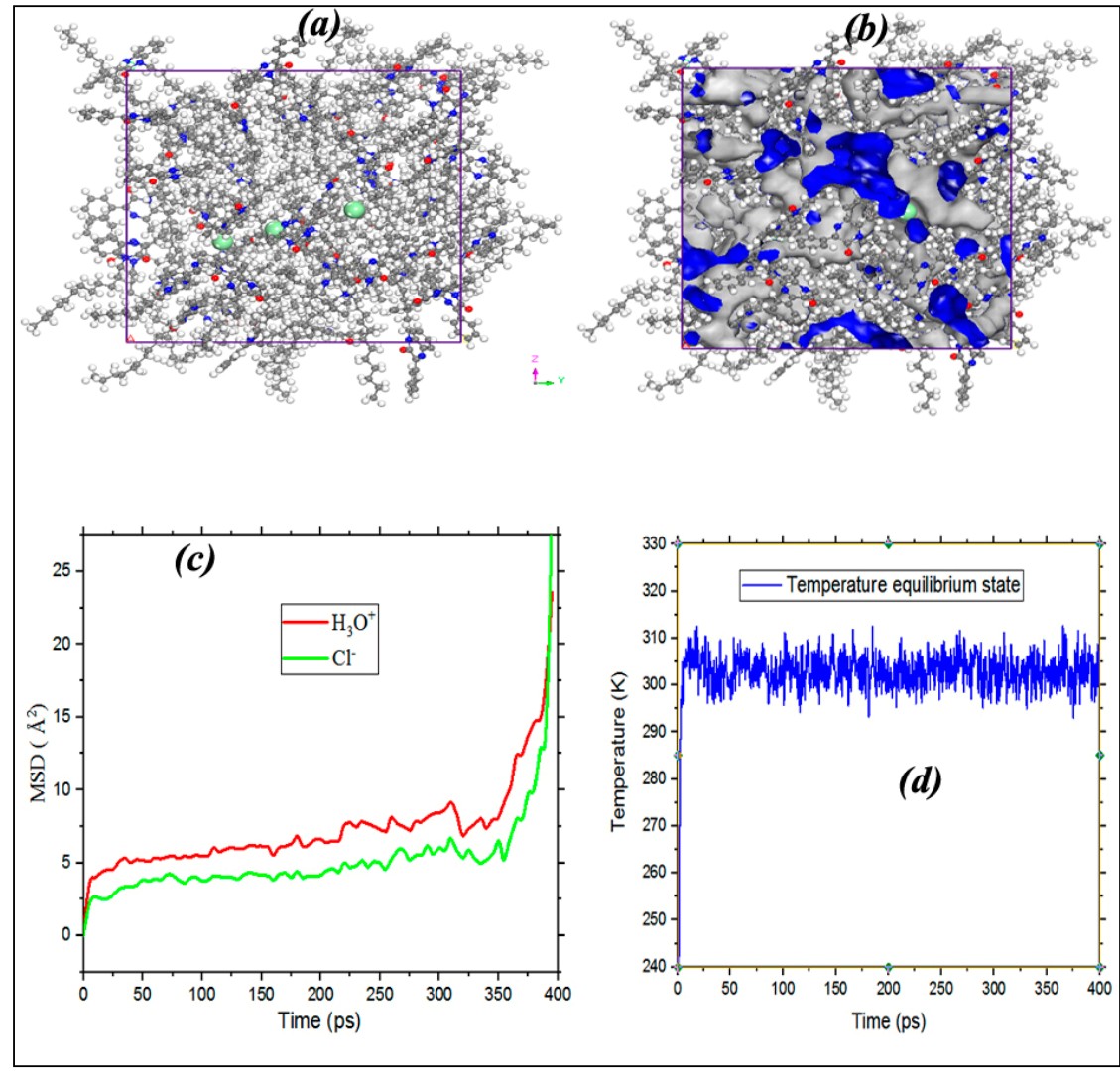

**Figure 15.** (**a**) Condensed QO12 films, (**b**) Connolly area, (**c**) MSD, and (**d**) temperature equilibrium state of Cl⁻ and H$_3$O⁺ in the QO12 films.

Figure 15b served to calculate the free Vf and occupied Vo volumes for the systems constructed with a Connolly area as obtained in the previous published research [73].

It is noteworthy that the higher FFV results from the presence of a larger vacant Vf inside the molecule film, and consequently the lesser effect of the corrosion-molecule film on the spread of corrosive elements [74]. In the present research, it was found that the FFV value of QO12 films corresponded to 21.08 %. This value is low, reflecting the success of the QO12 film in controlling the ingress of aggressive elements. This observation is also supported by the presence of the film which is formed from denser QO12 (Figure 15a).

In particular, the mean square displacement (MSD) in question measured the anticorrosion performance of 60 molecules of QO12 inside a supercell containing $3H_3O^+$ and $3Cl^-$ using Forcite with Materials Studio/8 software. The calculation of the diffusion coefficient ($D_{ion}$) can be performed using Equations (12) and (13) [72]:

$$D_{ion} = \frac{1}{6N_\alpha} \lim_{x \to \infty} \frac{d}{dt} \sum_{i=1}^{N_\alpha} \left\langle [R_i(t) - R_i(0)]^2 \right\rangle \tag{12}$$

$$MSD = \left\langle [R_i(t) - R_i(0)]^2 \right\rangle \tag{13}$$

Figure 15c reproduces the *MSD* graphs for the ions considered within the QO12 films. It is generally considered that if the $D_{ion}$ value is low, then the degree of inhibitory potency is high [75]. The values found for Figure 14c are $0.01795 \pm 0.00264 \ 10^{-12} \ m^2 \ s^{-1}$ for $Cl^-$ and $0.02098 \pm 0.00198 \ 10^{-12} \ m^2 \ s^{-1}$ for $H_3O^+$.

## 4. Conclusions

The anti-corrosive ability and inhibition mechanism of a novel quinoxalinone (QO12) against CS corrosion in an acidic electrolyte were investigated using systematic methodologies and characterization tools. The following conclusions may be formed based on the findings:

- QO12 has excellent effectiveness inhibiting CS corrosion in an acidic electrolyte and its performance improves as the amount rises, reaching a max of 95.33%, at $10^{-3}$ M. This efficiency value is closer to that of Q1 and Q2 in Table 1, while it is lower than that of the Q3 inhibitor, due to the presence of the nitro ($NO_2$) attracting group.
- The PDP profiles show that QO12 significantly inhibits anodic metal dissolution and cathodic hydrogen evolution processes, indicating that it is a mixed-type inhibitor with a cathodic tendency. EIS assessments show that the presence of the QO12 increases Rp values while decreasing the constant phase element of the double layer ($C_{dl}$), hence validating the inhibitor's inhibitory impact on CS corrosion.
- The chemisorption mechanism of QO12 adsorption on the CS interface is consistent with the Langmuir adsorption isotherm.
- Surface and electrolyte analyses (SEM, EDX, and UV-visible) suggest QO12 adsorption on the CS interface.
- Theoretical approaches indicate a good adsorption of QO12 on the selected surface.

**Author Contributions:** Conceptualization, F.B. and R.H.; methodology, M.M.; software, F.B.; validation, A.B., A.L. and Y.R.; formal analysis, S.L.; investigation, F.B.; resources, I.W.; data curation, H.O.; writing—original draft preparation, F.B.; writing—review and editing, F.B.; visualization, A.B.; supervision, A.Z.; funding acquisition, A.Z. All authors have read and agreed to the published version of the manuscript.

**Funding:** This research received no external funding.

**Institutional Review Board Statement:** Not applicable.

**Informed Consent Statement:** Not applicable.

**Data Availability Statement:** All data that support the findings of this study are included within the article.

**Conflicts of Interest:** The authors declare that they have no known competing financial interests or personal relationships that could have appeared to influence the work reported in this paper.

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
