# Peer review of "Theoretical and Experimental Studies of 1-Dodecyl-3-phenylquinoxalin-2(1H)-one as a Sustainable Corrosion Inhibitor for Carbon Steel in Acidic Electrolyte"

_coatings, doi:10.3390/coatings13061109_

Round 1

Reviewer 1 Report

Corrosion inhibitor method is a useful and meaningful technique for corrosion control. This work investigated the anti-corrosion feature and mechanism of QO12 and the experimental data was abundant. Some revisions need to be done before publication, as below:

1.      Background about the selection of QO12 needs to be added,

2.      Some grammar and spelling errors need to be corrected, like line 71, 161, 191, ect.

3.      What does the accuracy of electrochemical test depend on?

4.      Abbreviations need to be defined in this work, like PCs in line 87,χ² in Table 3. What’s the meaning of χ²? Relevant latest work may be useful, like:

(1)    Liao, B., Ma, S., Zhang, S., Li, X., Quan, R., Wan, S., & Guo, X. (2023). Fructus cannabis protein extract powder as a green and high effective corrosion inhibitor for Q235 carbon steel in 1 M HCl solution. International Journal of Biological Macromolecules, 239, 124358.

(2)    Zhou, Z., Min, X., Wan, S., Liu, J., Liao, B., & Guo, X. (2023). A novel green corrosion inhibitor extracted from waste feverfew root for carbon steel in H2SO4 solution. Results in Engineering, 17, 100971.

5.      Number of digit for the numerical value of β in Table 2 is wrong.

6.      To display the superior anti-corrosion performance, authors can compare the corrosion inhibition efficiency of QO12 with other reported organic corrosion inhibitors.

Some minor revisions need to be done before publication

Author Response

Response to Comments of Reviewers

We are pleased to inform you that based on the reviews; we have prepared the revised

Manuscript ID: coatings-2422758 R1and the rebuttal letter as a sign of positive response to the comments of your respected reviewers. We have carefully implemented all comments to improve the quality of our revised manuscript. To make it easier for you and reviewers, we have indicated the changes, corrections and additions in red color. In response to reviewer’s comments letter, we reproduced each comment by heading “Reviewer’s comment” and our responses have been given by heading “Author’s response”. We would like your final decision on our revised document for possible publication in the journal of coating in the special issue “Theoretical and Experimental Assessments of Corrosion Inhibitors”.

We appreciate your consideration of our manuscript, and we look forward to hearing from you soon. Any further information and suggestions are greatly appreciated.

Thank you very much for your consideration.

With best regards, Yours sincerely,

Dr. Fouad BENHIBA

Response to Comments of Editor and Reviewers:

 Reviewer 1

Corrosion inhibitor method is a useful and meaningful technique for corrosion control. This work investigated the anti-corrosion feature and mechanism of QO12 and the experimental data was abundant. Some revisions need to be done before publication, as below:

  1. Background about the selection of QO12 needs to be added,

Rep 1: The choice of inhibitor is added to the introduction and 2.1. Inhibitor structure section.

Thanks for this assessment

  1. Some grammar and spelling errors need to be corrected, like line 71, 161, 191, ect.

Rep 2: Thank you very much for these very important comments.

  1. What does the accuracy of electrochemical test depend on?

Rep 3:

The accuracy of an electrochemical test depends on several factors. Here are some key factors that can influence the accuracy of the test:

  • Calibration: Proper calibration of the electrochemical setup
  • Electrode Preparation: The preparation of electrodes is critical to obtaining accurate results.
  • Temperature Control: Temperature affects the kinetics of electrochemical reactions, so precise temperature control is important for accurate measurements
  • Electrolyte Composition and Concentration
  • Experimental conditions are necessary to minimize the impact on the measurements.
  • The number of tests for each concentration, typically 3.
  1. Abbreviations need to be defined in this work, like PCs in line 87,χ² in Table3. What’s the meaning of χ²? Relevant latest work may be useful, like:
  2.  

(1)    Liao, B., Ma, S., Zhang, S., Li, X., Quan, R., Wan, S., & Guo, X. (2023). Fructus cannabis protein extract powder as a green and high effective corrosion inhibitor for Q235 carbon steel in 1 M HCl solution. International Journal of Biological Macromolecules, 239, 124358.

(2)    Zhou, Z., Min, X., Wan, S., Liu, J., Liao, B., & Guo, X. (2023). A novel green corrosion inhibitor extracted from waste feverfew root for carbon steel in H2SO4 solution. Results in Engineering, 17, 100971.

Rep 4: Thank you for this improvement, we have taken these comments into account, see the manuscript

  1. Number of digit for the numerical value of β in Table 2 is wrong.

Rep 6: Thanks for your comment; we have homogenized the values of the Tafel slopes.

  1. To display the superior anti-corrosion performance, authors can compare the corrosion inhibition efficiency of QO12 with other reported organic corrosion inhibitors.

Rep 6: Thanks for your comment;  Table 1 summarizes some of the work carried out in literature.

Reviewer 2 Report

1.      Figure 8 lacks a specified unit on the y-axis, which should be included for better comprehension.

2.      The reference style employed in the manuscript does not conform to the prescribed journal style. Verify and utilize the appropriate reference style as recommended by the journal.

3.      It is advisable to augment the manuscript with more recent references to ensure an up-to-date and comprehensive literature review. Consider incorporating recent references wherever applicable.

4.      In Figure 7 and line number 71, address potential type errors.

5.      In Figure 6(b-d), the fitting does not align with the data points, despite the mentioned R2 value. It is crucial to provide a clear explanation and engage in a thorough discussion of these discrepancies within the Results and Discussion section.

6.      It is recommended to utilize schemes or figures consistently throughout the entire manuscript for improved visualization.

7.      While the author discusses the concentration of the inhibitor in decreasing order, it is important to consider the impact of higher concentrations, such as 10-2 M, and provide appropriate analysis.

8.      The author suggests exploring alternative acidic electrolytes for future research endeavors.

Author Response

Response to Comments of Reviewers

We are pleased to inform you that based on the reviews; we have prepared the revised

Manuscript ID: coatings-2422758 R1and the rebuttal letter as a sign of positive response to the comments of your respected reviewers. We have carefully implemented all comments to improve the quality of our revised manuscript. To make it easier for you and reviewers, we have indicated the changes, corrections and additions in red color. In response to reviewer’s comments letter, we reproduced each comment by heading “Reviewer’s comment” and our responses have been given by heading “Author’s response”. We would like your final decision on our revised document for possible publication in the journal of coating in the special issue “Theoretical and Experimental Assessments of Corrosion Inhibitors”.

We appreciate your consideration of our manuscript, and we look forward to hearing from you soon. Any further information and suggestions are greatly appreciated.

Thank you very much for your consideration.

With best regards, Yours sincerely,

Dr. Fouad BENHIBA

Response to Comments of Editor and Reviewer:

 Reviewer 2

  1. Figure 8 lacks a specified unit on the y-axis, which should be included for better comprehension.

Rep 1: Y-axis unit added, see figure 8.

  1. The reference style employed in the manuscript does not conform to the prescribed journal style. Verify and utilize the appropriate reference style as recommended by the journal.

Rep 2: Thank you for this important improvement, the recommendations are done accurately.

  1. It is advisable to augment the manuscript with more recent references to ensure an up-to-date and comprehensive literature review. Consider incorporating recent references wherever applicable.

Rep 3: The most recent references were added. Thank you for this improvement.

  1. In Figure 7 and line number 71, address potential type errors.

Rep 4: The error is rectified, thank you.

  1. In Figure 6(b-d), the fitting does not align with the data points, despite the mentioned R2 value. It is crucial to provide a clear explanation and engage in a thorough discussion of these discrepancies within the Results. and Discussion section.

Rep 5: Thanks for your suggestion, interpretation has been well added.

  1. It is recommended to utilize schemes or figures consistently throughout the entire manuscript for improved visualization.

Rep 6: Action taken – Thank you honorable reviewer

  1. While the author discusses the concentration of the inhibitor in decreasing order, it is important to consider the impact of higher concentrations, such as 10-2 M, and provide appropriate analysis.

Rep 7: The 10-2M concentration is more consumable, conversely, using corrosion inhibitors with low concentrations offers the benefit of minimizing costs associated with the inhibitor's procurement and application. Additionally, it can simplify handling and minimize the potential for unwanted side effects. However, it is crucial to ensure that the selected inhibitor and its concentration are appropriate for the specific corrosion situation to achieve the desired corrosion protection.

  1. The author suggests exploring alternative acidic electrolytes for future research endeavors.

Rep 8: Action taken – Thank you honorable reviewer

Reviewer 3 Report

This work investigates the corrosion inhibition efficiency of QO12. The author uses a comprehensive approach including electrochemical methods, surface characterization and DFT to obtain these results. Before this paper can be considered for publication, I would like the authors to address the following questions and comments.

1.     In general, although authors introduce the background in the introduction section, I feel they should add one paragraph to introduce QO12 before summarizing their results. Right now, the authors just mention QO12 directly without introducing some previous work about this material. Adding more details can help general readers to better understand the motivation/scope of this paper.

2.     In figure one, can authors add error bars to these data? This would be helpful to interpret the data.  

3.     For figure 8, I am curious what the SEM and EDX look like before immersion in HCI? This would be helpful to compare how bad the corrosion is without QO12.

4. In the conclusion, the authors summarize the main findings of the work, claiming that this new material can reach a max of 95.33 percent of inhibiting effectiveness. It would be good if they can compare this with the effectives of existing inhibitors (either here or in the introduction) to highlight the importance of QO12.

Author Response

Response to Comments of Reviewers

We are pleased to inform you that based on the reviews; we have prepared the revised

Manuscript ID: coatings-2422758 R1and the rebuttal letter as a sign of positive response to the comments of your respected reviewers. We have carefully implemented all comments to improve the quality of our revised manuscript. To make it easier for you and reviewers, we have indicated the changes, corrections and additions in red color. In response to reviewer’s comments letter, we reproduced each comment by heading “Reviewer’s comment” and our responses have been given by heading “Author’s response”. We would like your final decision on our revised document for possible publication in the journal of coating in the special issue “Theoretical and Experimental Assessments of Corrosion Inhibitors”.

We appreciate your consideration of our manuscript, and we look forward to hearing from you soon. Any further information and suggestions are greatly appreciated.

Thank you very much for your consideration.

With best regards, Yours sincerely,

Dr. Fouad BENHIBA

Response to Comments of Editor and Reviewers:

Reviewer 3

This work investigates the corrosion inhibition efficiency of QO12. The author uses a comprehensive approach including electrochemical methods, surface characterization and DFT to obtain these results. Before this paper can be considered for publication, I would like the authors to address the following questions and comments.

  1. In general, although authors introduce the background in the introduction section, I feel they should add one paragraph to introduce QO12 before summarizing their results. Right now, the authors just mention QO12 directly without introducing some previous work about this material. Adding more details can help general readers to better understand the motivation/scope of this paper.

Rep 1: Thanks for your suggestion, please see introduction section.

  1. In figure one, can authors add error bars to these data? This would be helpful to interpret the data.  

Rep 2: The error bars are added

  1. For figure 8, I am curious what the SEM and EDX look like before immersion in HCI? This would be helpful to compare how bad the corrosion is without QO12.

Rep 3: Thanks for your suggestion, please see introduction section, please see fig.8

  1. In the conclusion, the authors summarize the main findings of the work, claiming that this new material can reach a max of 95.33 percent of inhibiting effectiveness. It would be good if they can compare this with the effectives of existing inhibitors (either here or in the introduction) to highlight the importance of QO12.

Rep 4: Thanks for your suggestion, please see introduction section.

Reviewer 4 Report

The purpose of the present study was to investigate the behaviour of 1-dodecyl-3-phenylquinox-alin-2(1H)-one (QO12) and the corrosion inhibition on C.S. in 1M HCl solution. The corrosion inhibition effect of QO12 was evaluated using electrochemical techniques and surface characterisation. In addition, global quantum descriptor calculations using DFT and molecular dynamics simulation (MDS) were employed to gain a more comprehensive understanding of the experimental results. The research is interesting and combines experimental and theoretical studies. However, the manuscript itself needs improvement, the introduction is short and does not provide a sufficient overview of the research known so far or the selected molecule. In the experimental, more detailed descriptions should be given, especially about chemicals and processes. The discussion is weak, all results are strung together, and it is necessary to discuss at the end what the results show together and they should be commented on in a last short paragraph. Overall, the results are interesting, but the manuscript needs improvement. Some additional comments are listed below.

1.     The introduction is too short and does not provide background for the work as a whole, but in particular for the 1-dodecyl-3-phenylquinoxalin-2(1H)-one (QO12). You state in the conclusion that the molecule is novel: explain why in the introduction, why you chose it and what is the novelty of the research?

In the experimental part:

2.     It is not clear where or how the 1-dodecyl-3-phenylquinoxalin-2(1H)-one was obtained. Did you buy the chemical or was it synthesised, what was the preparation process?

3.     Sentence lines 62 and 63: The selection of 1-dodecyl-3-phenylquinoxalin-2(1H)-one (QO12) for use as a corrosion inhibitor was most likely based on its chemical structure and properties, and its ease of preparation and potential efficiency in preventing corrosion on steel surfaces. Most likely, seems like you are not sure why you chose this molecule or is this a literature citation?

4.     Your experiment only contains the instruments and methods. Include a small paragraph about the chemicals used, the manufacturers of the chemicals and solutions used, the concentrations, etc.

Results

5.     No standard deviation is given in the data presented (tables, figures) and should be included in the revised manuscript. Were the measurements conducted in replicates, how many, are the measurements reproducible?

Author Response

Response to Comments of Reviewers

We are pleased to inform you that based on the reviews; we have prepared the revised

Manuscript ID: coatings-2422758 R1and the rebuttal letter as a sign of positive response to the comments of your respected reviewers. We have carefully implemented all comments to improve the quality of our revised manuscript. To make it easier for you and reviewers, we have indicated the changes, corrections and additions in red color. In response to reviewer’s comments letter, we reproduced each comment by heading “Reviewer’s comment” and our responses have been given by heading “Author’s response”. We would like your final decision on our revised document for possible publication in the journal of coating in the special issue “Theoretical and Experimental Assessments of Corrosion Inhibitors”.

We appreciate your consideration of our manuscript, and we look forward to hearing from you soon. Any further information and suggestions are greatly appreciated.

Thank you very much for your consideration.

With best regards, Yours sincerely,

Dr. Fouad BENHIBA

Response to Comments of Editor and Reviewer:

 Reviewer 4

The purpose of the present study was to investigate the behaviour of 1-dodecyl-3-phenylquinox-alin-2(1H)-one (QO12) and the corrosion inhibition on C.S. in 1M HCl solution. The corrosion inhibition effect of QO12 was evaluated using electrochemical techniques and surface characterisation. In addition, global quantum descriptor calculations using DFT and molecular dynamics simulation (MDS) were employed to gain a more comprehensive understanding of the experimental results. The research is interesting and combines experimental and theoretical studies. However, the manuscript itself needs improvement, the introduction is short and does not provide a sufficient overview of the research known so far or the selected molecule. In the experimental, more detailed descriptions should be given, especially about chemicals and processes. The discussion is weak, all results are strung together, and it is necessary to discuss at the end what the results show together and they should be commented on in a last short paragraph. Overall, the results are interesting, but the manuscript needs improvement. Some additional comments are listed below.

Rep : Action taken – Thank you honorable reviewer

  1. The introduction is too short and does not provide background for the work as a whole, but in particular for the 1-dodecyl-3-phenylquinoxalin-2(1H)-one (QO12). You state in the conclusion that the molecule is novel: explain why in the introduction, why you chose it and what is the novelty of the research?

Rep 1: Action taken –please see introduction

In the experimental part:

  1. It is not clear where or how the 1-dodecyl-3-phenylquinoxalin-2(1H)-one was obtained. Did you buy the chemical or was it synthesised, what was the preparation process?

Rep 2: Action taken – the synthesis part is well inserted in the manuscript

  1. Sentence lines 62 and 63: The selection of 1-dodecyl-3-phenylquinoxalin-2(1H)-one (QO12) for use as a corrosion inhibitor was most likely based on its chemical structure and properties, and its ease of preparation and potential efficiency in preventing corrosion on steel surfaces. Most likely, seems like you are not sure why you chose this molecule or is this a literature citation?

Rep 3: QO12 used in this work is a new inhibitor synthesized, please see the synthesis part, and the sentence mentioned has been rectified. Thank you for this useful comment.

  1. Your experiment only contains the instruments and methods. Include a small paragraph about the chemicals used, the manufacturers of the chemicals and solutions used, the concentrations, etc.

Rep 4: Action taken –please see 2. Experimental section

Results

  1. No standard deviation is given in the data presented (tables, figures) and should be included in the revised manuscript. Were the measurements conducted in replicates, how many, are the measurements reproducible?

Rep 5: Action taken –Standard deviation is added in the data presented in all tables. All measurements were repeated three times for each experimental condition to ensure the reliability and reproducibility of the results, and the average values were noted.

Reviewer 5 Report

The article entitled " Theoretical and experimental studies of 1-dodecyl-3-phe- 2 nylquinoxalin-2(1H)-one as sustainable corrosion inhibitor for 3 carbon steel in acidic electrolyte”.  The paper focuses on the to study of the 1-dodecyl-3-phenylquinox- 52 alin-2(1H)-one (QO12) behavior, as well as the evaluation of the corrosion inhibition efficiency for carbon steel in 1M HCl solution, combined with quantochemical calculations. Although extensive work has been performed, several points must be improved before acceptance.

1.     The abbreviation "carbon steel should be made as it first appears in the abstract, page 1, line 19. I recommend noting it as CS without any points.

2.     Please clearly mention in the last paragraph of the introduction what the novelty of your work is, and what makes you different from other researchers.

3.     It is necessary to give more information about corrosion tests already mentioned in the literature on various compounds, to give a broader approach (use as references: https://doi.org/10.1515/pac-2018-0513, https://doi.org/10.1016/j.surfin.2020.100634, https://doi.org/10.1080/01496395.2017.1340953)

4.     Did you also try a saline solution for your experiments?

5.     At figures 1 and 4, Logi(mA cm-2) "i" must be written with a capital letter or to insert a space before i.

6.     I recommend using the same number of digits for the table values in Tables 2 and 3.

7.     References: Journal names need to be abbreviated. See https://www.mdpi.com/journal/coatings/instructions

Minor: Figure 1 do not appear in the main text.

               Be careful with subscript and superscript in the whole manuscript (ex. page 2, line 71)

                  Based on these, I advise the authors to rectify the above-mentioned issues, and I hope to re-evaluate the revised manuscript.

Author Response

-

Response to Comments of Reviewers

We are pleased to inform you that based on the reviews; we have prepared the revised

Manuscript ID: coatings-2422758 R1and the rebuttal letter as a sign of positive response to the comments of your respected reviewers. We have carefully implemented all comments to improve the quality of our revised manuscript. To make it easier for you and reviewers, we have indicated the changes, corrections and additions in red color. In response to reviewer’s comments letter, we reproduced each comment by heading “Reviewer’s comment” and our responses have been given by heading “Author’s response”. We would like your final decision on our revised document for possible publication in the journal of coating in the special issue “Theoretical and Experimental Assessments of Corrosion Inhibitors”.

We appreciate your consideration of our manuscript, and we look forward to hearing from you soon. Any further information and suggestions are greatly appreciated.

Thank you very much for your consideration.

With best regards, Yours sincerely,

Dr. Fouad BENHIBA

Response to Comments of Editor and Reviewer:

 Reviewer 5

The article entitled " Theoretical and experimental studies of 1-dodecyl-3-phe- 2 nylquinoxalin-2(1H)-one as sustainable corrosion inhibitor for 3 carbon steel in acidic electrolyte”.  The paper focuses on the to study of the 1-dodecyl-3-phenylquinox- 52 alin-2(1H)-one (QO12) behavior, as well as the evaluation of the corrosion inhibition efficiency for carbon steel in 1M HCl solution, combined with quantochemical calculations. Although extensive work has been performed, several points must be improved before acceptance.

  1. The abbreviation "carbon steel should be made as it first appears in the abstract, page 1, line 19. I recommend noting it as CS without any points.

Rep 1: Action taken –Thank you

  1. Please clearly mention in the last paragraph of the introduction what the novelty of your work is, and what makes you different from other researchers.

Rep 2: Action taken –please see introduction

  1. It is necessary to give more information about corrosion tests already mentioned in the literature on various compounds, to give a broader approach (use as references: https://doi.org/10.1515/pac-2018-0513, https://doi.org/10.1016/j.surfin.2020.100634, https://doi.org/10.1080/01496395.2017.1340953)

Rep 3: Action taken, thanks for your comment;  also Table 1 and Introduction summarize some of the work carried out in literature.

  1. Did you also try a saline solution for your experiments?

Rep 4: No, I will do this in future work.

  1. At figures 1 and 4, Logi(mA cm-2) "i" must be written with a capital letter or to insert a space before i.

Rep 5: Action taken –Thank you

  1. I recommend using the same number of digits for the table values in Tables 2 and 3.

Rep 5: Action taken –Thank you

  1. References: Journal names need to be abbreviated. See https://www.mdpi.com/journal/coatings/instructions

Minor: Figure 1 do not appear in the main text.

               Be careful with subscript and superscript in the whole manuscript (ex. page 2, line 71)

                  Based on these, I advise the authors to rectify the above-mentioned issues, and I hope to re-evaluate the revised manuscript.

Rep 7 : Action taken – Thank you honourable reviewer

Round 2

Reviewer 3 Report

i am satisfied with the modifications and now can happily recommend it for publication. 

Reviewer 4 Report

The purpose of the present study was to investigate the behaviour of novel 1-dodecyl-3-phenylquinox-alin-2(1H)-one (QO12) and the corrosion inhibition on C.S. in 1M HCl solution. The corrosion inhibition effect of QO12 was evaluated using electrochemical techniques and surface characterisation. In addition, global quantum descriptor calculations using DFT and molecular dynamics simulation (MDS) were employed to gain a more comprehensive understanding of the experimental results. The research is interesting and combines experimental and theoretical studies. The aim, design and methodology are consistently described. The authors took the comments into account and the manuscript is much more understandable and clearer. I recommend publishing the manuscript. 

Reviewer 5 Report

The authors have spent a lot of effort to further improve the manuscript, and they answered all of my questions well. Thus, I would recommend the Editor to consider an acceptance for publication in Coatings.